# A Randomized Trial of Tai Chi on Preventing Hypertension and Hyperlipidemia in Middle-Aged and Elderly Patients

**DOI:** 10.3390/ijerph18105480

**Published:** 2021-05-20

**Authors:** Jiansheng Wen, Min Su

**Affiliations:** Health Science Research Center of Tai Chi Association, Department of Physical Education, Northwestern Polytechnical University, Xi’an 710072, China

**Keywords:** cardiovascular disease, Tai Chi, the middle-aged and elderly

## Abstract

In our randomized controlled trial, we investigated whether Wu-style Tai Chi (Tai Chi combined with Daoyin) as a potential exercise prescription is more effective than simplified Tai Chi in the prevention and treatment of hypertension and hyperlipidemia in the middle-aged and elderly. We randomly assigned 66 patients with hypertension and hyperlipidemia to one of the two groups: the Wu-style Tai Chi group or the simplified Tai Chi group; the simplified Tai Chi group only exercised simplified Tai Chi three times a week for 6 weeks. The Wu-style Tai Chi group participated in 60 min of Wu-style Tai Chi three times a week for 6 weeks. Serum biochemical tests were conducted at baseline and at the end of the study. Measurements of blood pressure were performed at the same time. Primary outcomes were compared within and between groups at baseline and at 6 weeks. The participants in the Wu-style Tai Chi group performed, at 6 weeks, significantly better than baseline on all of the primary outcomes (*p* value ≤ 0.05). The results also show significant difference within the simplified Tai Chi group from baseline to 6 weeks in TCHO (mmol/L), SBP (mmHg), and LDL-C (mmol/L) (*p* value < 0.05). From baseline to 6 weeks, the Wu-style Tai Chi group had significant differences at more test indexes in serum and blood pressure than the simplified Tai Chi group. At 6 weeks, the Wu-style Tai Chi group had a significantly greater mean improvement in the SBP (mmHg) than did the simplified Tai Chi group (mean between-group difference, −5.80 (mmHg) [95% CI, −14.01 to 2.41]; *p* = 0.007). The results showed that, compared with simplified Tai Chi, Wu-style Tai Chi had a better effect on hypertension in the middle-aged and elderly. At 6 weeks in LDL-C (mmol/L), the Wu-style Tai Chi group had significantly greater improvement between the two groups (means between-group difference, −0.45 (mmol/L) [95% CI, −0.89 to −0.17]; *p* = 0.03). The results showed that Wu-style Tai Chi protected the cardiovascular system of the middle-aged and elderly in improving LDL-C (mmol/L), and was more significant than simplified Tai Chi. After 6 weeks of exercise, Wu-style Tai Chi could effectively improve hyperlipidemia and hypertension. The total effective rate of cardiovascular disease was 90.00%. There was significant difference in the treatment effect of hypertension and hyperlipidemia between the two groups during 6 weeks (*p* = 0.039), showing that, in a small population of middle-aged and elderly subjects, Wu style Tai Chi could be useful in managing important CV risk factors, such as hypertension and hyperlipidemia.

## 1. Introduction

Each year, 41 million people die from chronic diseases, which is equivalent to 71% of all deaths globally [1]. Chronic diseases will become the first burden of diseases among middle-aged and elderly people in the world. The main chronic diseases in middle-aged and old people are cardiovascular and cerebrovascular diseases [2]. Among them, cardiovascular disease is the number one killer of the health of middle-aged and elderly people, especially ischemic heart disease (IHD). IHD and ischemic stroke (IS) have become major causes of death in China, and the number of deaths is still on the rise [3].

Hypertension is an important risk factor for a variety of cardiovascular and cerebrovascular diseases, and with the increase of age, the prevalence of hypertension increases, and the risk of cardiovascular events increases significantly [4]. Serum LDL-C also plays an important role in the pathogenesis of atherosclerotic cardiovascular disease (ASCVD) [5].

It is another major risk factor for IHD and IS [6,7].

Hypertension, heart disease, cerebral thrombosis, arteriosclerosis, and other cardiovascular diseases have threatened the physical and mental health of middle-aged and elderly people all over the world for a long time. With the development of science and technology, people gradually realize the important role of Tai Chi exercise in health, prevention, and treatment of cardiovascular and cerebrovascular diseases.

Tai chi is a kind of fitness activity guided by the mind. It combines exercise with static state, and is naturally relaxing and calming. At the same time, it is accompanied by breathing exercises. It plays an important role in the prevention and treatment of chronic diseases in most middle-aged and elderly people. At present, there have been many studies on Tai Chi for cardiopulmonary rehabilitation training of middle-aged and elderly people [8,9].

Steven et al., through a 48-week clinical randomized experiment, found that the cardiovascular function of the Tai Chi exercise group was significantly enhanced when compared to the health education group [10]. Further research found that Tai Chi can improve the sleep quality of patients with chronic heart failure, and that it has certain benefits on blood pressure, arrhythmia, and one’s quality of life [11]. David and other scholars conducted more in-depth research. The results showed that the psychological stress levels of subjects receiving decompression course education still existed, and the level of nuclear factor of kappa B (NF-κB) significantly increased, while the subjects in the Tai Chi group showed that the effects of psychological pressure decreased and the NF-κB level remaining unchanged. This shows that practicing Tai Chi can relieve mental pressure and improve the quality of life [12].

NF-κB is a transcription factor that regulates the expression of various cellular genes, and is involved in the pathological process of coronary heart disease, such as atherosclerosis and myocardial ischemia. Inhibition of NF-κB activation is expected to open up a new way for the prevention and treatment of coronary heart disease. Tai chi can also improve other risk factors of cardiovascular disease, such as blood glucose and proteinuria (common in patients with hypertension), which can affect the prognosis of these patients [13].

According to a meta-analysis conducted by researchers from Shanghai Sport University in 2016, Tai Chi can benefit patients with cardiovascular disease, reduce blood pressure and blood lipid levels, and improve heart failure symptoms, physical strength, and mood [14]. However, compared with mature “medical means”, as an exercise prescription, Tai Chi has not been a mastered and targeted application.

Meta analyses showed that, in terms of exercise intervention measures, simplified Tai Chi was the most used exercise intervention means in the study of Tai Chi, and the intervention time used in the study was mostly concentrated in 12 weeks (60 min/time, 3 times/week). However, this evidence should be due to the limitations of the method [15].

In future research, there are several directions worthy of improvement, such as the extraction of effective movement elements, and getting rid of the limitations of simplified Tai Chi.

In this study, the blood pressure and blood lipids of the Wu-style Tai Chi group and simplified Tai Chi group were compared and observed. Wu style Tai Chi is used as a means of sports intervention. Compared with simplified Tai Chi, Wu style Tai Chi movements are softer and continuous. It is more suitable for the physiological characteristics of the cardiovascular system of middle-aged and elderly people. It belongs to the traditional Tai chi handed down from ancient China, which is different from simplified Tai chi created in 1956. In traditional Wu style Tai Chi, Daoyin (Combined with qigong and movement guidance, used in ancient Chinese medicine to treat diseases) is inherently included. This paper discusses whether Wu Style Tai Chi, as a potential exercise prescription, is more effective than simplified Tai Chi in the prevention and treatment of hypertension and hyperlipidemia in middle-aged and elderly people.

Sixty-six patients were randomly divided into two groups. The intervention time and weekly frequencies were the same: the simplified Tai Chi group (*n* = 33) 3 times a week, 60 min each time; the Wu-style Tai Chi group (*n* = 33) 3 times a week, 60 min each time; for a total of 6 weeks of observation. We analyzed the effects of two exercises on blood pressure and blood lipids of middle-aged and elderly people.

## 2. Methods

### 2.1. Study Design

We designed a randomized clinical trial to compare the effects of exercise at 6 weeks in the Wu-style Tai Chi group with the effects in the simplified Tai Chi group. Each group participated in a 60-min class that met 3 times weekly for 6 weeks. We did not use a double-blind study design because this would require fake Tai Chi, for which there is currently no proven method.

In the random cycle, we used computer-generated numbers to assign allocations to Wu-style Tai Chi or simplified groups. The research was carried out in accordance with the test protocol. The randomly assigned treatment tasks were sealed in opaque participants and opened separately for each patient who agreed to participate in the study. In order to ensure the comparability of baseline between groups and increase the credibility of the results, the gender factors can be balanced by dividing the gender layer first and then randomly assigning within each gender layer by the randomization method.

All subjects gave their informed consent for inclusion before they participated in the study. The study was conducted in accordance with the Declaration of Helsinki, and the protocol was approved by the Medical Ethics Committee of Northwestern Polytechnic University (the World Health Organization international clinical trials registered organization, registered platform Clinical Trials Registration number: ChiCTR2000040613. (http://www.chictr.org.cn/edit.aspx?pid=65296&htm=4, accessed on 3 December 2020). All authors vouch for the completeness and accuracy of the data and attest to the fidelity of the trial to the protocol. The results in this article have not been previously published. The Health Science Research Center of the Tai Chi Association generates a random allocation sequence, recruits participants, and allocates them to participate in the intervention.

### 2.2. Study Participants

Research participants recruited from three cities (Beijing, Xi’an, Hohhot) through the recommendations of professional doctors and the distribution of information to groups of patients with hypertension and hyperlipidemia.

After screening, subjects were randomly divided into the Wu-style Tai Chi group and simplified Tai Chi group, and both groups started and ended at the same time. The total observation period of the study subjects was 6 weeks.

#### 2.2.1. Eligibility Criteria Include

(1) Clinical diagnosis of hypertension and hyperlipidemia: we observed 66 patients with stage 1 to 3 disease, according to a disease staging scale, in two groups. In recent years, the medical community has formed a three-stage prevention program for early detection, early treatment, and prevention of chronic disease prevention and treatment.

Stage 1 prevention refers to a comprehensive control of multiple risk factors when no cardiovascular disease is formed clinically, i.e., source control; Stage 2 prevention refers to the use of medication or non-medication to improve chronic diseases in patients with cardiovascular disease; Stage 3 prevention refers to the rehabilitation of patients with cardiovascular disease, prevention or delay of the occurrence of chronic comorbidities, prevention of rehospitalization and death due to recurrence of chronic diseases [16,17,18].

(2) Judgment criteria: systolic blood pressure (SBP) >140 mmHg or/and diastolic blood pressure (DBP) >90 mmHg were hypertension, SBP 120~139 mmHg or/and DBP 80~89 mmHg were normal high blood pressure. TCHO ≥ 5.20 mmol/L or/and TG ≥ 1.76 mmol/L, LDL–C ≥ 3.37 mmol/L, HDL-C < 1.04 mmol/L is dyslipidemia.

(3) Between 40 to 75 years old.

(4) Had no regular physical exercise for at least 1 year (3 months with a frequency of 3 to 4 times per week and 30 min per session were considered as the minimal standard for regular physical exercise).

(5) Stable medication use or no medication.

#### 2.2.2. Exclusion Criteria Were

(1) Any history of stroke, severe cerebrovascular disease, musculoskeletal system disease, or sports injury related contraindications;

(2) The cognitive screening by the Mini-Mental State Exam [19].

### 2.3. Intervention

Wu-style Tai Chi group exercise is based on the movement of Wu-style Tai Chi and Daoyin, based on the theory of traditional Chinese medicine meridians, and the exercise combination is designed according to hypertension and hyperlipidemia. In this trial, we designed the combination of the classic Wu-style of Tai Chi and Daoyin for treating cardiovascular diseases. It is designed by Wu-style Tai Chi and Daoyin’s professional masters and professional traditional Chinese medicine doctors who have been teaching for 30 years. The simplified Tai Chi group adopted the usual 24 simplified Yang-style Tai Chi.

#### 2.3.1. Wu-Style Tai Chi Group

Wu-style Tai Chi intervention took place three times a week for 6 weeks, and each session lasted for 60 min. Classes were taught by a Tai Chi Daoyin master with more than 30 years of teaching experience. In the first session, he explained the theory behind Tai Chi Daoyin and its procedures and provided participants with printed materials on its principles and techniques. In subsequent sessions, participants practiced 60 forms from the classic Wu style of Tai Chi and Daoyin for treating cardiovascular diseases under his instruction. Each session included a warm-up and self-massage, followed by a review of principles, movements, breathing techniques, and relaxation in Wu-style Tai Chi. Throughout the intervention period, participants were instructed to practice Wu-style Tai Chi at home for at least 30 min each day. At the end of the 6-week intervention, participants were encouraged to maintain their Tai Chi practice, using an instructional DVD, Keep the follow-up visit.

#### 2.3.2. Simplified Tai Chi Group

At the beginning of the trial, the simplified Tai Chi group was taught a simplified Tai Chi course. Classes were taught by a Tai Chi Daoyin master with more than 30 years of teaching experience. Simplified Tai Chi intervention took place three times a week for 6 weeks, and each session lasted for 60 min. Throughout the intervention period, participants were instructed to practice simplified Tai Chi at home for at least 30 min each day.

Two groups were asked to maintain their regular drug use within 6 weeks of the trial.

### 2.4. Adherence to Programs

During the nine-week intervention, participants in both groups continued their daily activities normally, but were asked not to participate in any new additional exercise program. During the entire 6-week intervention, the researchers asked the participants who missed the course to make up missed lessons. We recorded the attendance of each course and asked the participants to fill out a questionnaire form indicating the time they were practicing Tai Chi.

### 2.5. Outcome Measures

Weight determination: body weight and height were measured, and BMI was calculated. The standard of judgement: BMI > 24.0 kg/m^2^ is overweight, BMI > 28.0 kg/m^2^ is obesity. There was no significant difference in age, sex, and body mass index between the two groups (all *p* > 0.05). The setting and location of collecting data is Northwestern Polytechnic University Hospital.

#### 2.5.1. Primary Outcomes

The primary outcome measure was the change in blood pressure, blood lipids in the blood routine from baseline to the end of the 6-week intervention.

The blood pressure was measured by a desktop sphygmomanometer. Moreover, 30 min of rest before blood pressure was measured, 3 times continuously. We took the mean and count according to WHO diagnostic criteria for hypertension. Blood pressure measurement: according to the blood pressure measurement method recommended by China’s hypertension prevention and treatment guidelines, a routine sleeve type mercury sphygmomanometer was used by the full-time nurses to measure the right brachial artery pressure for the subjects in the sitting position.

Blood lipid and blood glucose measurement: fasting blood sampling, samples were 3 mL of fasting venous blood. Total cholesterol (TC), triglyceride (TG), serum high density lipoprotein cholesterol (HDL-C), and serum low density lipoprotein cholesterol (LDL-C) were tested by the full automatic biochemical analyzer.

#### 2.5.2. Secondary Outcomes

Effective percentage of Wu-style Tai Chi in the treatment of chronic diseases. It refers to the rate of improvement or significant improvement and cure of chronic diseases in the Wu-style Tai Chi group during the Wu-style Tai Chi intervention.

Special physical measurements were conducted on each case of chronic disease in the Wu-style Tai Chi group before and after 6 weeks of treatment. Typical physiological indexes were detected and a special color Doppler ultrasound, type-B ultrasonic analysis, chest X-ray fluoroscopy, and electrocardiogram were performed.

Throughout the entire intervention period, we monitored adverse events, using a standard adverse-event case report form at each visit. This form included a description of all unanticipated benefits and undesirable experiences [18].

#### 2.5.3. Clinical Treatment Effect Criteria

Effective percentage of cardiovascular diseases. It refers to the rate of improvement and cure of cardiovascular diseases in two groups during the observation period. Special physical measurements were conducted in each case of cardiovascular diseases in two groups before and after 9 weeks of treatment.

Significant improvement: diastolic blood pressure (DBP) and systolic blood pressure (SBP) reach the normal range; the level of fasting blood lipid reaches the normal range. In medication use, participants stopped taking the medicine.

Become better: although the systolic or diastolic blood pressure did not fall to normal, the systolic and diastolic blood pressure decreased by 10 mmHg or more. Although the range of fasting blood lipid level did not fall to normal, the range of TCHO decreased by 0.5 mmol/L or more, and the range of TG decreased by 0.3 mmol/L or more. At the same time, the range of HDL-C increased by 0.2 mmol/L or more and the range of LDL-C decreased by 0.2 mmol/L or more. In medication use, participants reduced the medicine.

### 2.6. Statistical Analysis

Statistical analysis was performed using SPSS 26.0 Software (SPSS Inc., Chicago, IL, USA). A threshold of *p* < 0.05 (2-tailed) was applied.

Between-group differences in demographic and baseline variables were tested with a chi-square test for categorical variables and a one-way analysis of variance for continuous variables. One-way ANOVA and chi-square tests were used to analyze the baseline demographic characteristics between groups.

Paired *t*-tests were used to examine within group changes from baseline to 6 weeks. Independent-sample *t*-tests (with 95% confidence intervals) were used to compare group means at baseline.

Because the subjects of the study have different chronic diseases, the trend of blood indicators before intervention is not the same. This will mask the effects of the intervention during statistical analysis. Therefore, covariance analysis was used to analyze the difference between the two groups at 6 weeks after testing the adjusted covariates. A two-sided *p* value of less than 0.05 indicated statistical significance. we randomly assigned 66 patients to two groups (33 patients to each), which provided 78% power to detect a difference between means at a significance level of 5% with the use of a two-sided *t*-test.

## 3. Results

### 3.1. Baseline Characteristics of the Participants

From December 2020 through January 2021, a total of 126 persons were screened for eligibility. Moreover, 76 qualified for the baseline evaluation and 10 patients in this group were excluded for various reasons, and the 66 eligible participants were randomly assigned in equal numbers to either Wu-style Tai Chi intervention or simplified Tai Chi intervention (Figure 1). Six patients withdrew from the study by 6 weeks (Figure 1).

Table 1 shows the baseline characteristics of the study population. There were no significant differences between the two groups in age, gender, the mean body mass index (the weight in kilograms divided by the square of the height in meters), average years of education, duration of chronic disease, and chronic disease stages at the beginning of the treatment (*p* > 0.05). Baseline characteristics were reasonably well balanced between the two groups. The average attendance rate in the Wu-style Tai Chi group was 89%, ranging from72% to 100%, while in the simplified Tai Chi it was 88%, ranging from 71% to 100% (*p* value > 0.05). Due to various reasons, 3 participants in each group dropped out of the trial, and finally 30 participants in each group completed the experiment. Excluding the data of dropouts, the results of the trial were calculated according to the data of each group of 30 participants who completed the trial. There were no significant differences in the baseline demographic variables between participants who completed the trial.

### 3.2. Outcomes

Mean (±SD) within-group differences in outcomes at Baseline and 6 weeks are shown in Table 2. *p* values were calculated with paired *t*-tests. Paired *t*-tests were used to examine within group changes from baseline to 6 weeks. The participants in the Wu-style Tai Chi group performed, at 6 weeks, significantly better than baseline on all the primary outcomes (*p* value ≤ 0.05). Table 2 also shows significant difference within the simplified Tai Chi group from baseline to 6 weeks in TCHO (mmol/L), SBP (mmHg), LDL-C (mmol/L) (*p* value < 0.05). Table 2 shows no significant difference in DBP (mmHg), HDL-C (mmol/L) and TG (mmol/L) (NS; *p* value > 0.05) from baseline to 6 weeks within the simplified Tai Chi groups. From baseline to 6 weeks, the Wu-style Tai Chi group had significant difference at more test indexes in serum and blood pressure than the simplified Tai Chi group.

Means (±SD) within group difference (95% CI) in outcomes at 6 weeks are shown in Table 2. The measurement means of the Wu-style Tai Chi group are closer to the normal value, and the standard deviation fluctuation range is smaller than that of the simplified Tai Chi group.

The difference within the two groups at 6 weeks was significant than the baseline in TCHO (mmol/L), SBP (mmHg), LDL-C (mmol/L), (*p* value < 0.05), indicating that Wu-style Tai Chi and simplified Tai Chi are all effective for treating cardiovascular disease.

The results show that Wu style Tai Chi has a more comprehensive treatment and better effect in improving cardiovascular diseases than simplified Tai Chi. For further confirmation, a statistical comparison between groups was conducted.

Table 3 shows changes at baseline and 6 weeks between the two groups for all outcomes.

There was no significant difference in any baseline biochemical parameters between the two groups, as shown in Table 3.

At 6 weeks, the Wu-style Tai Chi group had a significantly greater mean improvement in the SBP (mmHg) than did the simplified Tai Chi group (mean between-group difference, −5.80 (mmHg) [95% CI, −14.01 to 2.41]; *p* = 0.007). The results showed that compared with simplified Tai Chi, Wu-style Tai Chi had a better effect on hypertension in the middle-aged and elderly.

At 6 weeks in LDL-C (mmol/L), the Wu-style Tai Chi group also had significantly greater improvement between the two groups (means between-group difference, −0.45 (mmol/L) [95% CI, −0.89 to −0.17]; *p* = 0.03). The results showed that Wu-style Tai Chi, by protecting the cardiovascular system of the middle-aged and elderly, by improving LDL-C (mmol/L), was more significant than simplified Tai Chi.

In addition, the Wu-style Tai Chi group had greater improvement in TCHO (mmol/L), DBP (mmHg), HDL-C (mmol/L), and TG (mmol/L) at 6 weeks. However, the difference was not significant (*p* value > 0.05).

This is primarily because, at 6 weeks, the simplified Tai Chi group also had improvement in some biochemical parameters. At the same time, as shown in Table 3, the Wu-style Tai Chi group had greater improvement in the SBP (mmHg) and LDL-C (mmol/L).

The results further show that Wu-style Tai Chi exercise therapy had a more comprehensive treatment and better effect than simplified Tai Chi in improving middle-aged and elderly patients with cardiovascular disease during the same intervention period.

Low density lipoprotein cholesterol: low density lipoprotein cholesterol (LDL-C) is about 2.7 mmol/L (105 mg/dL) in young adults and 3.1 mmol/L (120 mg/dL) in middle-aged and elderly adults. If LDL-C is above 4.14 mmol/L, it indicates that LDL-C is on the high side. ‖ Triglyceride: The normal range is: male 0.44~1.76 mmol/L; female 0.39~1.49mmol/L. High-density lipoprotein cholesterol: male < 40 years old 0.9~1.83 mmol/L (30~59 mg/dL), female < 40 years old 1.1~2.0 mmol/L (33~77 mg/dL); ‖ triglyceride: the normal range is: male 0.44~1.76 mmol/L; female 0.39~1.49 mmol/L.

Table 4 shows cardiovascular disease clinically meaningful improvement between Wu-style Tai Chi group and simplified Tai Chi group during 6 weeks. In medication use, at 6 weeks, more exercisers discontinued medication used to treat cardiovascular disease in the Wu-style Tai Chi group than in the simplified Tai Chi group. Nine patients in the Wu-style Tai Chi group stopped taking the medicine (significant improvement), 18 reduced the medicine (become better), and 3 did not change the medicine. In the simplified Tai Chi group, 8 stopped taking the medicine (significant improvement), 17 reduced the medicine (become better), and 5 did not change the medicine. *p* values = 0.039, intervention effects remained significant after adjustment baseline chronic disease stages varying covariates.

Total effective percentage was 90.00% in Wu-style Tai Chi group. Total effective percentage was 73.33% in the simplified Tai Chi group. Table 4 shows that Wu-style Tai Chi can significantly improve cardiovascular disease. No adverse events were noted during the study period.

There was significant difference in the treatment effect of hypertension and hyperlipidemia between the two groups during 6 weeks (*p* = 0.039). The results showed that Wu-style Tai Chi had better clinically meaningful improvement in hypertension and hyperlipidemia than simplified Tai Chi.

## 4. Discussion

Human aging is a process of physiological development. With age, due to the loss of blood vessel elasticity, calcium deposition and glial fiber proliferation in the blood vessel wall, the dilation of blood vessels is reduced, and progressive arteriosclerosis appears, which increases peripheral resistance, blood pressure, and left ventricular overload, and causes left ventricular insufficiency, which together leads to many cardiovascular diseases.

The movements of Tai Chi are soft and continuous, which is in line with the psychological and physiological characteristics of middle-aged and old people. Studies have found that Tai Chi exercise has many positive effects on the health of middle-aged and older people.

This study shows the simplified Tai Chi group has improvement in some biochemical parameters. Our observation results are consistent with the previous research results of Tai Chi in the prevention and treatment of cardiovascular disease [14].

However, this study also shows that Wu-style Tai Chi is potentially better effective therapy for patients with cardiovascular disease than simplified Tai Chi. Compared with the simplified Tai Chi group, Wu-style Tai Chi exercise not only reduced the level of SBP (mmHg) leading to cardiovascular disease, but also significantly reduced the level of LDL-C (mmol/L) in patients with cardiovascular disease. In the blood pressure and blood lipids tests, improvements were significant. No adverse events were observed during Tai Chi exercise, indicating the safety and usefulness of this exercise for persons with cardiovascular disease. After 6 weeks of Wu-style Tai Chi exercise, Wu-style Tai Chi could effectively improve hyperlipidemia and hypertension. Total effective rate of cardiovascular disease was 90.00%. There was significant difference in the treatment effect of hypertension and hyperlipidemia between the two groups during 6 weeks (*p* = 0.039), showing that Wu-style Tai Chi had better clinically meaningful improvement in hypertension and hyperlipidemia than simplified Tai Chi.

From 1997 to 2020, the worldwide study of Tai Chi developed steadily, and the number of published papers showed an increasing trend. Tai chi to improve balance ability, fibromyalgia and posture control is research field that is being explored [20,21,22]. Interventions for chronic diseases, such as cardiovascular disease and osteoarthritis, are at the forefront of Tai Chi research [23,24,25,26,27]. This research is clear, gradually deepening, and is formed on the basis of Tai Chi’s research on sports-related diseases. Tai chi research is increasingly being widely accepted in the medical field.

By systematically observing and testing blood pressure changes of Tai Chi practitioners who practice Tai Chi throughout the year, and non-trainers, before and after completion of quantitative load exercise—the heart rate and blood pressure recovery speed of the Tai Chi exercise group has a tendency to increase compared with the control group after increasing the exercise load. This shows that Tai Chi exercise can improve the adaptability of the vascular function of the middle-aged and elderly people, and enhance athletic ability. Therefore, long-term Tai Chi exercise cannot only improve the elasticity of the blood vessel wall of the elderly, but also improve the exercise function of the elderly and prevent hypertension [28,29].

The connection of all the movements of traditional Wu style Tai Chi is created according to the rhythm of reverse abdominal breathing. The rhythm of breathing is matched with gentle movements. The reverse abdominal breathing can provide more return heart blood volume and stroke volume. This helps circulatory metabolism. Simplified Tai Chi was newly compiled in 1956, with the purpose of popularizing Tai Chi and reducing the difficulty of memory. On the basis of traditional Yang style Tai Chi, many movements are reduced, and 24 movements are selected. The connection of these 24 movements does not correspond to reverse abdominal breathing. This will relatively reduce the impact of Tai Chi on the cardiovascular system [30,31].

In previous exercise intervention measures for cardiovascular diseases, simplified Tai Chi is the most used exercise intervention method in the research on Tai Chi, and the intervention time used in the research mainly focuses on 12 weeks (60 min/each time, 3 times/week) [21,32,33,34,35]. According to the results of this study, to improve the patient’s physical condition, the appropriate type of Tai Chi should be selected according to the patient’s disease type, and the Wu-style Tai Chi practice should be selected to improve blood pressure and blood lipid status. For cardiovascular disease risk factors, exercise prescriptions are formulated according to the patient’s physical condition and disease type to achieve the best treatment.

This study has some limitations.

First, the main limitation of the study is the modest number of patients recruited into the study. Being a lifestyle intervention trial (Wu-style Tai Chi vs. simplified Tai Chi), it would have been necessary to define a simple size and an adequate duration of the trial that would allow defining the durability of the effectiveness of a type of exercise with respect to the exercise intervention.

Second, we only observed for 6 weeks. The long-term efficacy of Tai Chi Daoyin for patients with cardiovascular disease remains to be determined. There are many types of Tai Chi, and how to choose and effectively treat cardiovascular disease is still in the exploratory stage.

In addition, a proportion of the subjects recruited in the study were elderly. Furthermore, in both groups studied, the prevalence of the two genders was not the same. Disease distribution showed that malignancy, diabetes, coronary artery disease, chronic kidney disease, and chronic obstructive pulmonary disease were more frequent in men, but hypertension, osteoarthritis, anemia, and depression were more frequent in women [36].

Experimental grouping can be considered stratified based on age and sex.

A meta-analysis published in the *European Journal of Cardiovascular Nursing* summarized 15 previous clinical studies (13 randomized controlled trials, 2 quasi trials) on Tai Chi, including patients with coronary heart disease, chronic heart failure, hypertension, and stroke. The duration of exercise intervention in these studies varied from 6 to 52 weeks (average 17 weeks) [37]. Another meta-analysis reviewed 20 scientific literatures on Tai Chi as an exercise intervention for the prevention and treatment of heart disease.

## 5. Conclusions

These studies suggest that Tai Chi is a safe form of exercise for the prevention and treatment of cardiovascular disease (CVD). However, the sample sizes were generally small, with only 20% of the studies having 100 or more participants [38].

In terms of outcome indicators, Tai Chi practice has significant effects on improving blood pressure and blood lipids. However, further studies with more rigorous designs, larger sample sizes, more adequate exercise amounts of Tai Chi, and stricter selections of measurement criteria to evaluate the mechanisms and effects of Tai Chi are needed, before the medical effects of Tai Chi can be widely disseminated.

In conclusion, in a small population of middle-aged and elderly subjects, Wu style Tai Chi could be useful in the management of vital CV risk factors, such as hypertension and hyperlipidemia. Longer-term studies involving larger clinical samples are necessary to assess the universality of our findings and to deepen our understanding of this promising treatment.

## Figures and Tables

**Figure 1 ijerph-18-05480-f001:**
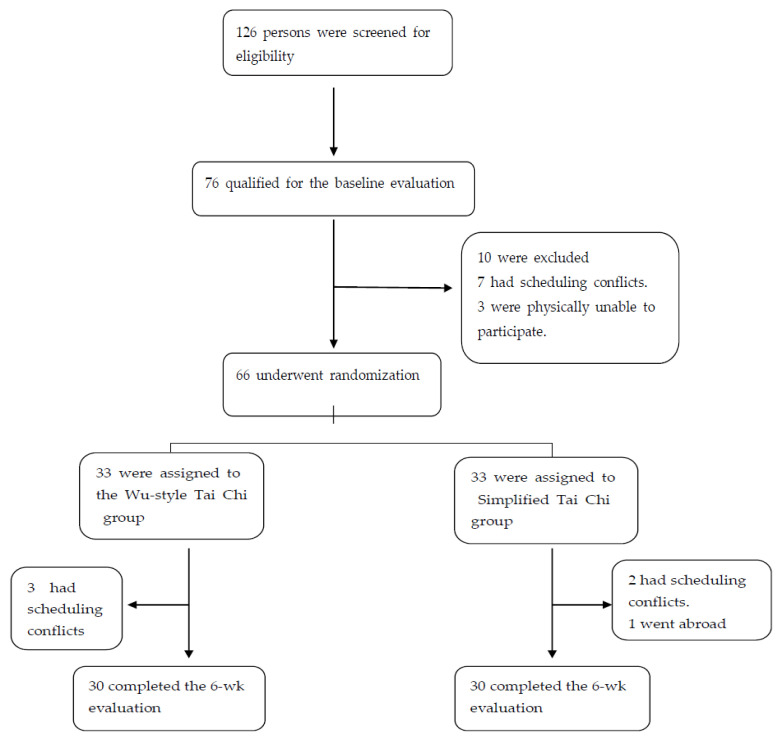
Screening, randomization, and completion of baseline and 6-week evaluations.

**Table 1 ijerph-18-05480-t001:** Demographic and clinical characteristics of the study participants at baseline *.

	Wu-Style Tai Chi	Simplified Tai Chi	*p* Value
Characteristic	(*n* = 30)	(*n* = 30)	
Gender (female/male)	12/18	12/18	1.000
Age-year	57.77 ± 9.46	57.20 ± 9.00	0.813
Education Years ^†^	9.33 ± 2.84	9.03 ± 2.50	0.666
Body mass index ^‡^	23.51 ± 2.32	23.83 ± 2.40	0.604
Duration of disease—y	3.92 ± 1.99	3.80 ± 2.05	0.824
Chronic disease stages (no./*n*) ^§^			0.953
Stage 1	11/30	12/30	
Stage 2	10/30	9/30	
Stage 3	9/30	9/30	
Self-reported coexisting number of chronic diseases	1.63 ± 1.13	1.73 ± 1.11	0.731

* Plus–minus values are means ±SD with the 95% confidence intervals. The chi-square test was used for categorical variables, and one-way analysis of variance for continuous variables. There were no significant between-group differences in any baseline characteristics. ^†^ High school or higher education; ^‡^ The body mass index is the weight in kilograms divided by the square of the height in meters; ^§^ We randomized 66 patients with stage 1 to 3 disease on a staging scale to one of the following two groups. In recent years, the medical community has formed a three-stage prevention program for early detection, early treatment, and prevention of chronic disease prevention and treatment.

**Table 2 ijerph-18-05480-t002:** Study measures, differences within groups at baseline and 6 weeks *.

	Within-Group Differences (95% CI) ^†^		
Variable	Mean ± SD (95% CI)		
	Baseline	Week 6	*p* Value ^†^
TCHO (mmol/L) ^‡^			
Simplified Tai Chi (*n* = 30)	5.17 ± 1.09	4.90 ± 1.09	0.012
Wu-style Tai Chi (*n* = 30)	5.02 ± 1.20	4.54 ± 0.79	<0.001
SBP (mmHg) ^§^			
Simplified Tai Chi (*n* = 30)	134.30 ± 20.81	124.30 ± 19.01	0.009
Wu-style Tai Chi (*n* = 30)	136.83 ± 19.91	123.50 ± 11.98	<0.001
DBP (mmHg) ^§^			
Simplified Tai Chi (*n* = 30)	80.26 ± 13.43	77.90 ± 12.78	NS
Wu-style Tai Chi (*n* = 30)	80.83 ± 13.16	76.30 ± 8.81	0.004
LDL-C (mmol/L)			
Simplified Tai Chi (*n* = 30)	3.27 ± 0.98	2.94 ± 0.96	<0.001
Wu-style Tai Chi (*n* = 30)	3.01 ± 1.01	2.48 ± 0.71	<0.001
HDL-C(mmol/L)			
Simplified Tai Chi (*n* = 30)	1.36 ± 0.35	1.36 ± 0.42	NS
Wu-style Tai Chi (*n* = 30)	1.40 ± 0.34	1.51 ± 0.27	0.036
TG (mmol/L) ^‖^			
Simplified Tai Chi (*n* = 30)	1.70 ± 0.81	1.64 ± 0.79	NS
Wu-style Tai Chi (*n* = 30)	1.73 ± 0.63	1.52 ± 0.47	0.050

* Plus–minus values are means ± SD, with the 95% confidence intervals. NS denotes not significant. ^†^
*p* values were calculated with paired *t*-tests. Paired *t*-tests were used to examine within group changes from baseline to 6 weeks. ^‡^ Total cholesterol, normal value 3–5.2 mmol/L or so. ^§^ The normal blood pressure range is: systolic blood pressure (SP) 140–90 mmHg, diastolic blood pressure (DP) 90–60 mmHg. ^‖^ Low density lipoprotein cholesterol: low density lipoprotein cholesterol (LDL-C) is about 2.7 mmol/L (105 mg/dL) in young adults and 3.1 mmol/L (120 mg/dL) in middle-aged and elderly adults. If LDL-C is above 4.14 mmol/L, it indicates that LDL-C is on the high side. ^‖^ Triglyceride: The normal range is: male 0.44~1.76 mmol/L; female 0.39~1.49 mmol/L. High-density lipoprotein cholesterol: male <40 years old 0.9~1.83 mmol/L (30~59 mg/dL), female < 40 years old 1.1~2.0 mmol/L (33~77 mg/dL); ^‖^ triglyceride: the normal range is: male 0.44~1.76 mmol/L; female 0.39~1.49 mmol/L.

**Table 3 ijerph-18-05480-t003:** Study measures between-group differences in the change at baseline and 6 weeks *.

	Between-Group Mean Difference (95% CI) ^†^			
Variable	Wu-Style Tai Chi	vs.	Simplified Tai Chi	*p* Value
	(*n* = 30)		(*n* = 30)	
TCHO (mmol/L) ^‡^				
Baseline	−0.15 (−0.76	to	0.44)	NS
Week 6	−0.36 (−0.85	to	0.13)	NS
SBP (mmHg) ^§^				
Baseline	2.53 (−7.99	to	13.06)	NS
Week 6	−5.80 (−14.01	to	2.41)	0.007
DBP (mmHg) ^§^				
Baseline	0.57 (−6.31	to	7.44)	NS
Week 6	−1.60 (−7.29	to	4.09)	NS
LDL-C (mmol/L)				
Baseline	−0.26 (−.77	to	−0.26)	NS
Week 6	−0.45 (−0.89	to	−0.17)	0.03
HDL-C (mmol/L)				
Baseline	0.04 (−0.14	to	0.21)	NS
Week 6	0.15 (−0.03	to	0.33)	NS
TG (mmol/L) ^‖^				
Baseline	−0.03 (−0.34	to	0.41)	NS
Week 6	−0.12 (−0.46	to	0.21)	NS

* Plus–minus values are means ± SD, with the 95% confidence intervals. NS denotes not significant. Independent-sample *t*-tests (with 95% confidence intervals) were used to compare group means at Baseline. ^†^ The values shown are unadjusted means with the 95% confidence intervals. *p* values were calculated with analysis of covariance. Point estimates and estimates falling within the 95% confidence interval were generated from independent *t*-tests for group differences. Because the subjects of the study have different chronic diseases, the trend of blood indicators before intervention is not the same. This will mask the effects of the intervention during statistical analysis. Therefore, covariance analysis was used to analyze the difference between the two groups at 6 weeks after testing the adjusted covariates. A two-sided *p* value of less than 0.05 indicated statistical significance. ^‡^ Total cholesterol, Normal value 3–5.2 mmol/L or so. ^§^ The normal blood pressure range is: systolic blood pressure (SP) 140–90 mmHg, diastolic blood pressure (DP) 90–60 mmHg. ^‖^ Triglyceride: The normal range is: male 0.44~1.76 mmol/L; female 0.39~1.49 mmol/L. High-density lipoprotein cholesterol: male <40 years old 0.9~1.83 mmol/L (30~59 mg/dL), female <40 years old 1.1~2.0 mmol/L (33~77 mg/dL); triglyceride: the normal range is: male 0.44~1.76 mmol/L; female 0.39~1.49 mmol/L.

**Table 4 ijerph-18-05480-t004:** Cardiovascular disease clinically meaningful improvement between the Wu-style Tai Chi group and the simplified Tai Chi group during 6 weeks *.

	Wu-Style Tai Chi	Simplified Tai Chi	*p* Value
	(*n* = 30)	(*n* = 30)	
Cardiovascular disease			
(Hypertension ^†^ + Hyperlipidemia ^‡^)			0.039
medication use (Cases)			
No change	3	8	
Become better *	18	17	
Significant improvement *	9	5	
Total effective percentage (%) ^§^	90.00	73.33	

* Because the subjects of the study have different chronic diseases, the trend of blood indicators before intervention is not the same. This will mask the effects of the intervention during statistical analysis. Therefore, covariance analysis was used to analyze the difference between the two groups at 6 weeks after testing the adjusted covariates. A two-sided *p* value of less than 0.05 indicated statistical significance. According to the diagnostic criteria of the World Health Organization (WHO). See clinical treatment effect criteria for criteria of significant improvement and become better in outcome measures section. ^†^ Hypertension is a clinical syndrome characterized by elevated systemic arterial blood pressure (systolic blood pressure (>140 mm Hg) and diastolic blood pressure (>90 mm Hg). ^‡^ Hyperlipidemia refers to high blood lipid levels, which can directly cause diseases that seriously endanger human health, such as atherosclerosis, coronary heart disease, pancreatitis, etc., fasting TCHO, TG, LDL-C, HDL-C. ^§^ Become better + Significant improvement/total cases (%).

## Data Availability

The data presented in this study are available in Appendix A.

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
