# Peer review of "A Randomized Trial of Tai Chi on Preventing Hypertension and Hyperlipidemia in Middle-Aged and Elderly Patients"

_ijerph, 2021, doi:10.3390/ijerph18105480_

Round 1

Reviewer 1 Report

Tai Chi is a well-developed martial art and exercise which has great potential to improve some chronic diseases. In this manuscript, the authors try to evaluate the benefit of Tai Chi for hypertension and hyperlipidemia. The authors designed two different Tai Chi forms for the trial and found both Wu-style Tia Chi and simplified Tai Chi improved hypertension and hyperlipidemia in a 6-week duration. The authors also found Wu-style Tia Chi has more beneficial than simplified Tai Chi. The topic is very interesting, and the results are very promising. There are some items to be addressed.

  1. The number of patients in the final statistics is confusing. In the last paragraph of the introduction, the authors said Sixty patients were randomly divided into 2 groups, however, both the n of both Wu-style and simplified Tai Chi groups are 33. In table 1, n=30, but the female/male ratio is 13/20.
  2. All the items in table 1 are matched so well, it is not like randomly assigned participants in each group. It is normal to match the important items to minimize the influence of some factors in clinical trials. The method should be recorded accordingly.
  3. Table 1, the total number of patients for the Chronic disease stages in Simplified Tai Chi only 29 patients. However, there should be 30 participants in this group. How about the patient left there?
  4. In figure 1, the words in the figures were separated. It is hard to read. The authors should reschedule the figure to make the word fit in the same line.
  5. Why does Wu-style Tai Chi benefit more to patients than simplified Tai Chi? The authors should explain in the discussion section.

Author Response

Dear Reviewer: 

   Thank you for your letter and for the reviewer's comments concerning our manuscript entitled “A randomized trial of Tai Chi on preventing hypertension and hyperlipidemia in middle-aged and elderly patients ” ( ID: ijerph-1215913).       

   Those comments are all valuable and very helpful for revising and improving our paper, as well as the important guiding significance to our researches. We have studied comments carefully and have made correction which we hope meet with approval. The main revised portion are marked in red in the paper. The main corrections in the paper and the responds to the reviewer's comments are as flowing:

Reviewer Comments: Reviewer 1Tai Chi is a well-developed martial art and exercise which has great potential to improve some chronic diseases. In this manuscript, the authors try to evaluate the benefit of Tai Chi for hypertension and hyperlipidemia. The authors designed two different Tai Chi forms for the trial and found both Wu-style Tia Chi and simplified Tai Chi improved hypertension and hyperlipidemia in a 6-week duration. The authors also found Wu-style Tia Chi has more beneficial than simplified Tai Chi. The topic is very interesting, and the results are very promising. There are some items to be addressed.

1).Response to comment: (The number of patients in the final statistics is confusing. In the last paragraph of the introduction, the authors said Sixty patients were randomly divided into 2 groups, however, both the n of both Wu-style and simplified Tai Chi groups are 33. In table 1, n=30, but the female/male ratio is 13/20. )

Response:

We'd like to thank the reviewer for careful readings and valuable comments.

Because at the beginning of the trial, there were 66 people randomized, 33 in each group.

Due to various reasons, 3 subjects in each group dropped out of the experiment, and finally 30 subjects in each group completed the experiment. Excluding the data of dropouts, the results of the experiment were calculated according to the data of each group of 30 subjects who completed the experiment. Due to falling off (cannot be avoided during the experiment). There were different numbers at the beginning and the end, so there was some confusion. Thank you for correcting me.

We have revised Table 1 and added a note before Table 1 (see the part highlighted in red). All data were counted according to the final results of each group of 30 people. See Table 1 (see red). Relevant parts of The Introduction have been corrected (see the part marked in red).We have revised Table 1 and added a note before Table 1 (see the part highlighted in red). All data were counted according to the final results of each group of 30 people. See Table 1 (see red). Relevant parts of The Introduction have been corrected (see the part marked in red). 2). Response to comment: (All the items in table 1 are matched so well, it is not like randomly assigned participants in each group. It is normal to match the important items to minimize the influence of some factors in clinical trials. The method should be recorded accordingly.)

Response:

We'd like to thank the reviewer for careful readings and valuable comments.

The male/female sex ratio is because we applied stratified randomization, and the specific method is added in the method. In order to ensure the comparability of baseline between groups and increase the credibility of the results, the gender factors can be balanced by dividing the gender layer first and then randomly assigning within each gender layer by randomization method.See the red part of 2.1. Study designBefore entering the group, we have screening criteria, we analyze each group randomly to form such a match, which should be related to the screening at the time of entering the group.

3). Response to comment: (Table 1, the total number of patients for the Chronic disease stages in Simplified Tai Chi only 29 patients. However, there should be 30 participants in this group. How about the patient left there?)

Response:

We'd like to thank the reviewer for careful readings and valuable comments.

This is indeed an oversight. In the simplified Tai Chi group, there is an elderly patient in the rehabilitation stage. We have included her in the Chi-square test. However, she was ignored in the stage of the disease recorded in Table 1. We revised table 1 again, see table 1Her blood biochemistry and blood pressure were fully measured during the experiment.

4). Response to comment: (In figure 1, the words in the figures were separated. It is hard to read. The authors should reschedule the figure to make the word fit in the same line.)

Response:

We'd like to thank the reviewer for careful readings and valuable comments.

It has been corrected according to your comments, as shown in Figure 1 

5). Response to comment: (Why does Wu-style Tai Chi benefit more to patients than simplified Tai Chi? The authors should explain in the discussion section.)

Response:

We'd like to thank the reviewer for careful readings and valuable comments.

Thank you for your suggestion, it has been added to the discussion section. The connection of all the movements of traditional Wu style Tai Chi is created according to the rhythm of reverse abdominal breathing. The rhythm of breathing is matched with gentle movements. The reverse abdominal breathing can provide more return heart blood volume and stroke volume. This all helps circulatory metabolism. Simplified Tai Chi was newly compiled in 1956 with the purpose of popularizing Tai Chi and reducing the difficulty of memory. On the basis of traditional Yang style Tai Chi, many movements are reduced, and 24 movements are selected. The connection of these 24 movements does not correspond to reverse abdominal breathing. This will relatively reduce the impact of Tai Chi on the cardiovascular system.

References: Jiansheng Wen, Min Su. "study on the physiological effect of Tai Chi reverse abdominal breathing." Journal of Beijing Sport University 35.03 (2012): 67-70 doi:10.19582/j.cnki.11-3785/g8.2012.03.014.

Liufan Zhang. The effect of different frequencies of forward and reverse abdominal breathing on the effect of cranial blood supply to healthy male college students. 2018. Beijing Sport University, MA thesis.

See the part of the discussion marked in red.

We tried our best to improve the manuscript and made changes in the manuscript.  These changes will not influence the content and framework of the paper. And the changes marked in red in revised paper.

We appreciate for Reviewer's warm work earnestly, and hope that the correction will meet with approval.

Once again, thank you very much for your comments and suggestions.

Reviewer 2 Report

The manuscript: „ A randomized trial of Tai Chi on preventing hypertension and hyperlipidemia in middle-aged and elderly patients„ by Jiansheng Wen and Min Su analyzes the effectiveness of Wu-style Tai Chi over simplified Tai Chi in the prevention and treatment of hypertension and hyperlipidemia. Based on the outcome of the study, the authors conclude that the Wu-style Tai Chi is clinically effective than Simplified Tai Chi in terms of improvement of Hypertension and Hyperlipidemia. After going through the manuscript, I have following comments:

  1. Please describe the basic differences between Wu-style Tai Chi and Simplified Tai Chi.
  2. The introduction section is too lengthy and clumsy. Please consider trimming the introduction. The detailed methodology of referred studies (for example Page 2, paragraph 6; Steven et al. randomly divided………….) is not required.
  3. The participants were recruited from three different cities. It seems that the participants started the exercise at the same time. However, there might be some operational discrepancies in performing the exercises. Different tutors have their own style of teaching the exercises. How was it assured that there was no or very little discrepancy in the exercise methodology in participants in different cities?
  4. Please mention how uniform was the diagnosis of hypertension and hyperlipidemia in patients in different cities.
  5. Was sample size sufficient to derive the conclusions based on statistical analyses?
  6. There was just marginal statistical difference (p=0.039) in treatment effect of hypertension and hyperlipidemia between two groups after the trial period. This fringe difference could have arisen due to methodological discrepancies. Please provide your arguments in favor of deriving the conclusions based on this result.
  7. Please double check the total number of participants in Wu-style Tai Chi and simplified Tai Chi (both n=30) in table 1. The Gender distribution shows 33 participants in each group (female/male: 13/20 in each group).
  8. Were there any differences/similarities in different parameters at baseline and week 6 based on gender?
  9. Both the groups seem to be perfectly age matched based on means of the ages of participants. How was the median age scenario?
  10. The manuscript needs to be scrutinized for grammatical errors and typos. Hence a proof reading preferably by a native English speaker is recommended.

Reviewer 3 Report

The paper could be interesting for the readers. However, this reviewer raises several criticisms that have to be addressed by the Authors.

1- The main limitation of the study is the modest number of patients recruited into the study. Being a lifestyle intervention trial (Wu-style Tai Chi vs Simplified Tai Chi), it would have been necessary to define a simple size and an adequate duration of the trial that would allow to define the durability of the effectiveness of a type of exercise with respect to the 'eltro. This aspect must be fully commented in the discussion and also in the limitations section.

2- In the conclusions the authors write 'our preliminary results demonstrate that Wu style Tai Chi is a useful treatment for Cardiovascular disease.' This concept is also in the abstract. This claim is not supported by the results of the study. Actually, the authors can just state that, on a small population of middle-aged and elderly subjects, Wu style Tai Chi could be useful in the management of important CV risk factors, such as hypertension and hyperlipidemia.

3- The authors evaluated the impact of Tai Chi only on some CV risk factors, such as hypertension and hyperlipidemia. On such a small number of patients it would have been useful and easy to evaluate the effect of Tai Chi on other important CV risk factors such as glycaemia and proteinuria (very frequent in hypertensive patients) which impact the outcome of these patients (Nephrol Dial Transplant. 2018 Nov 1;33(11):1942-1949.  doi: 10.1093/ndt/gfy032). A comment on this issue and this reference should be included in the text.

4- A proportion of the subjects recruited in the study are elderly. Furthermore, in both groups studied, the prevalence of the two genders is not the same. In the elderly population, gender affects morbidity and outcome as observed in the REPOSI study (Eur J Intern Med. 2014 Sep; 25 (7): 617-23. doi: 10.1016/j.ejim.2014.06.027). This issue and this study should be commented upon.

5- All tables, especially 2 and 3, and figure must be re-formatted, as they are not well readable in the submitted file.

6- The manuscript requires the linguistic revision of a native English speaker.

Round 2

Reviewer 3 Report

None